# Navigating a New Normal: A Qualitative Look at Long-Term Care Planning for Children with Intellectual Disabilities Post-COVID-19

**DOI:** 10.3390/healthcare12242512

**Published:** 2024-12-11

**Authors:** Alice Yip, Yuen-Han Mo, Jeff Yip, Zoe Tsui, Fu-Fai Fong, Pui-Man Chu

**Affiliations:** 1S.K. Yee School of Health Sciences, Saint Francis University, 2 Chui Ling Lane, Tseung Kwan O, New Territories, Hong Kong, China; ztsui@sfu.edu.hk; 2Department of Social Work, Hong Kong Shue Yan University, 10 Wai Tsui Crescent, Braemar Hill, North Point, Hong Kong, China; yhmo@hksyu.edu (Y.-H.M.); fffong@hksyu.edu (F.-F.F.); 3Hong Kong Institute of Paramedicine, Hong Kong, China; jeffreyyip@gmail.com; 4Hong Chi Winifred Mary Cheung Morninghope School, 220 Lai King Hill Road, Kwai Chung, Hong Kong, China; wmc@hongchi.org.hk

**Keywords:** COVID-19, family cares, intellectual disabilities, long-term care, qualitative research

## Abstract

**Background:** Family caregivers of individuals with intellectual disabilities (ID) face numerous challenges in long-term planning, which have been exacerbated amidst the COVID-19 pandemic. Specific triggers raise awareness of future planning needs, but barriers like painful emotions and exhaustion often impede the process. This study aimed to explore Hong Kong (HK) caregivers’ perspectives on long-term planning for family members with ID at the later period of the pandemic. **Methods:** A qualitative phenomenological approach was utilized. In-depth interviews regarding experiences with long-term care preparation during COVID-19 were conducted with 12 purposively sampled HK caregivers of adults with ID. Data were analyzed using Colaizzi’s method. **Results:** Four key themes emerged: planning a loved one’s future alone, the burden of arranging care for disabled loved ones, planning a child’s future care amid family tensions, and the pandemic worsening future caregiving worries. **Conclusion:** Caregivers urgently require encouragement and support from policymakers and professionals to build confidence in long-term strategy and access robust assistance. Implications include identifying caregiver concerns, aiding gradual planning implementation, increasing respite options, and facilitating discussions regarding future residential care homes. This study provides initial valuable insights into an overlooked population during an unprecedented crisis.

## 1. Introduction

Family caregivers of individuals with intellectual disabilities (ID) who require extensive care and assistance may need to arrange temporary alternative caregiving arrangements due to factors like caregiver illness, declining health associated with advanced age, burnout from caregiving responsibilities, or the combination of multiple caregiving duties [1]. Securing substitute caregivers to take over for a significant duration helps provide respite and prevent interruption of essential support for the individual with a disability when primary caregivers are unable to continue providing care. Individuals with intellectual and developmental disabilities may reside in a range of domiciles, including private family homes, community-based group homes, supervised residential facilities, or other supported living environments. Their family member caregivers—whether parents, siblings, spouses/partners, or other relatives—often provide care and assistance for the individual with disability irrespective of their housing situation or residential model. Securing alternative caregiving arrangements allows families a respite when primary caregivers are temporarily unable to provide the intensive support required. Parents and siblings of people with ID frequently experience high levels of unease and concern about the future [2,3]. This stems from a lack of robust arrangements for others to take over caregiving responsibilities when they are no longer able. Crafting comprehensive future plans that detail contingency options can substantially allay the disquiet and trepidation that parents and siblings feel. Having clear alternatives mapped out in advance provides relief by assuring them that their family member will continue receiving needed support. Most research shows that families rarely make and record clear, specific plans for who will care for individuals with ID when parents and siblings are no longer able to [4]. While family members often have optimistic visions for who will provide future care when they no longer can, studies indicate these desired arrangements frequently stay unspoken rather than being directly negotiated and settled upon with those envisioned caregivers [5,6].

Studies detail some possible advantages for family caregivers of relatives with disabilities. Reported positives include an enhanced sense of meaning and significance in their lives, greater composure and equanimity in difficult times, as well as more gratitude for simple daily pleasures. Providing ongoing assistance to disabled family members can spur personal evolution and shifts in values that caregivers regard as emotionally enriching. The act of caring deeply for kin and committing to their welfare can reshape caregivers’ philosophies in uplifting ways. Research describes how dedicated support of disabled family members can bring about a favorable change in caregivers’ worldviews and perspectives [7]. Though some research describes possible benefits such as increased meaning, composure, and appreciation of small joys when caring for relatives with intellectual disabilities, the preponderance of evidence stresses that this role still levies major burdens regarding the physical, emotional, financial, and time demands put on family member caregivers. Numerous studies reveal that relatives of those with ID frequently endure adverse physical and mental health effects [8,9,10]. Research documents elevated levels of strain, distress, sadness, and other impacts among these caregivers [11]. Having a sibling with cognitive impairment can profoundly affect family dynamics and have lasting impacts on siblings’ well-being [11,12]. The body of evidence makes clear that growing up alongside those with ID can impose substantial burdens on relatives across physical, psychological, and social realms.

The increased life expectancy of individuals with ID makes proactive long-term care planning on the part of their parents an urgent priority, as the growing likelihood that those with ID will survive their parental caregivers due to the rising longevity highlights the emerging necessity for parents to develop comprehensive long-term care strategies that will provide support for their children beyond the parents’ own lifetimes [13,14]. As parents of children with ID grow older, they become more worried about continuing care for their children after the parents die. This concern makes sense because older parents face more health problems while still trying to plan lifelong support for their intellectually disabled child. Making plans for care after they pass away is also hard for parents because their disabled children are living longer lives. This means parents need to find caregivers who can help their children for many years. The situation shows why older parents urgently need to make future care plans to reduce their worries and make sure their disabled children will be looked after when the parents’ health fails completely. Moreover, external support infrastructure is deficient for a preponderance of parental caregivers of intellectually disabled progeny, imposing upon them an isolated and ceaseless care burden replete with exhaustive obligations requisite to providing for their child’s extensive needs [5]. As siblings of children with ID grow up, they start worrying that they will have to take on more caregiving for their intellectually impaired brother or sister [15,16]. This added pressure on siblings, who feel obligated to help more as adults, makes things even harder on parents who are already stressed caring for a disabled child. Some research indicates siblings experience dual responsibilities and heightened pressures when they must provide caregiving for an intellectually impaired sibling in addition to assisting their parents and tending to their own nuclear family [17]. This compounds difficulties across the family system.

Family caregivers of the ID often avoid planning for the future because of many obstacles, like mistrusting care facilities and painful emotions of giving up care duties and facing their eventual death [18]. To ensure their disabled relative’s well-being, caregivers need to overcome these impediments and make responsible plans for when family members can no longer provide care. Caregivers also face obstacles like uncertainty on how to initiate planning, the reluctance to overburden others, and the lack of capable people in their lives to hand over care duties to, highlighting the need for guidance on launching the process and building a supportive care network [5,11]. Also, people in the caregiver’s life, like healthcare providers usually do not talk about or bring up making future care plans [5,6]. Also, intellectually impaired relatives can sometimes create problems for future planning, like adamantly refusing to depart their household. This may be because their disability makes it hard for them to be adaptable or handle changes to their routine, environments, and caregivers.

The worldwide COVID-19 contagion may increase awareness among people with ID and their families of the need to make plans and develop long-term strategies. This global health emergency could encourage individuals, organizations, and governments to rethink how they help prepare people with ID for future emergencies and unexpected situations. When COVID-19 reached Hong Kong (HK), most families faced challenges from the pandemic. Residential care homes for people with ID in HK prohibited visitors for over 6 months during the initial pandemic response in order to prevent viral transmission. With HK affected by the COVID-19 crisis, daycare centers, schools, and other facilities completely closed for over 6 months to prevent vital transmission. Home care for people with ID continued at times but was disrupted by containment, infection, and fear of spreading the infection. In general, family members of people with ID received less expert assistance and relief during the pandemic as daycare centers, schools, and healthcare services were closed or limited for over 6 months [19,20].

The COVID-19 pandemic and protective actions like quarantine have presented additional burdens and difficulties for caregivers of individuals with ID [21,22]. The COVID-19 pandemic created major disruptions for families through the closure of childcare facilities and schools, reduced social contact, and interruptions to routines, constituting an exceptionally challenging event for family well-being [23,24]. This overwhelming situation posed unique challenges, especially for vulnerable households, including those facing socioeconomic adversity, caregiver mental health struggles, or children with special needs [25,26]. For these disadvantaged families, the pandemic’s wide-ranging disruptions likely constituted an exceptionally difficult experience.

The pandemic presented them with uncertain and risky circumstances for their well-being and existence, which frequently resulted in involvement in future preparations. The COVID-19 pandemic has exacerbated the challenges faced by isolated family caregivers who have potentially needed to escalate their caregiving obligations, intensifying their duties and leading them to recognize both their own susceptibility and the importance of distributing responsibilities throughout their caregiving community [20,27,28]. In addition to isolation, visiting limitations elicited concerns regarding the welfare of family caregivers and their ID relatives, the exchange of information between them, and the role of familial support compared to assistance from healthcare experts [29,30].

Nevertheless, the perspectives of family caregivers providing support for their ID relatives in HK have not yet undergone systematic investigation, and limited research exists regarding family caregivers of individuals with ID. The extent to which inferences from Western societies are relevant to family caregivers in HK continues to be uncertain, given the city’s singular mix of Eastern and Western cultural mindsets. Investigating the perspectives of family caregivers in HK could prove informative for illuminating the challenges they encounter, such that their experiences may guide and enlighten service planning and advancement. The study focuses on illuminating the experiences of family caregivers in HK caring for their adult family members with ID, with the goal of gaining insight into caregivers’ perspectives on the community-based services they have utilized. The investigation aims to explore HK caregivers’ perspectives on long-term planning for family members with ID at the later period of the pandemic.

## 2. Methods

### 2.1. Study Design

This qualitative study used in-depth interviews to explore the experiences of family caregivers caring for family members with ID during the latter period of the COVID-19 pandemic. The data were analyzed using Colaizzi’s phenomenological methodology [31]. This approach aims to comprehend participants’ subjective perspectives and lived experiences by mentally reconstructing the phenomenon under investigation. Through phenomenological analysis, the researchers sought to illuminate the essence of caregiving amidst the pandemic from the caregivers’ point of view.

### 2.2. Participants

The purposive sample comprised nine HK family caregivers of adults with ID who were educated in special schools till data saturation. The inclusion criteria were: (i) being the family caregivers of a child with ID who had completed or was soon to complete the secondary school curriculum in HK; (ii) being fluent in the Cantonese language; and (iii) living with a child with an ID.

### 2.3. Access to Participants

The researchers (A.Y. and Y.-H.M.) carried out all interviews from March 2022 to September 2023. Interviews were conducted at each participant’s convenience either via Zoom (using a private, password-protected meeting), telephone, or in person (Nasdaq, San Jose, CA, USA). The interviews took place in a one-on-one format between the researcher and the participant. The interviews lasted approximately 80 to 100 min. To promote consistency during the data-gathering process, the researcher utilized the same set of interview questions for each participant encounter. To confirm the validity of the interview questions, a panel of two experts—one a scholar who specialized in qualitative research and the other a psychological consulting specialist—evaluated the interview questions (Appendix A). With the participants’ written consent, the researcher audio-recorded all the interviews for later review. The researchers took field notes during each interview session. This allowed for the recording of contextual information to supplement the audio recordings and support the subsequent data analysis process. The complete set of interview guide questions was developed by the researchers. During the interview, the researcher monitored for emotional distress among participants. If any distress was evident, appropriate psychological support was provided to avoid further harm.

### 2.4. Analysis Strategy

The individual interview recordings were transcribed word-for-word by two researchers (A.Y. and J.Y.) utilizing NVivo software (NVivo Version 12, QSR International, Melbourne, Australia). To methodically analyze the transcriptions and field notes, Colaizzi’s phenomenological approach was employed [31]. This involved a seven-step process of initial data familiarization led to the identification of significant verbatim meaning units, largely through emergent coding. Researchers then interpreted the meaning of each unit, bracketing their biases, and grouped similar units into initial codes. This involved both emergent and intentional coding, iteratively refining a preliminary codebook through researcher discussion. These codes were synthesized into a comprehensive description of participants’ experiences, which was then analyzed to identify overarching themes and subthemes (primarily an intentional process). To ensure accuracy and participant authenticity, transcriptions were provided to participants for verification and approval—a process known as member checking. Finally, the researchers validated these themes by comparing them to the original data, ensuring the final thematic structure reflected a synthesis of both emergent and intentional coding approaches.

### 2.5. Rigor

Colaizzi’s phenomenological analysis was applied to each interview transcription [31]. Three research team members (A.Y., J.Y., and Z.T.) separately evaluated the transcripts, summarized meaningful statements, and identified themes, as shown in Table 1. Comparing and discussing subthemes enabled spotting similarities and differences entirely. Rigor was ensured by applying Lincoln and Guba’s criteria for credibility, dependability, confirmability, and transferability [32]. Member checking and reaching consensus further validated the analysis. To enhance the quality and transparency of reporting, this study followed the Consolidated Criteria for Reporting Qualitative Research (COREQ) checklist [33].

### 2.6. Ethical Consideration

The study received ethical approval from the Research and Ethics Committee of a local university (HREC 22-05(M07)). Participants were fully informed of the study’s purpose and methodology and provided written consent. Participation was voluntary, confidentiality was maintained through de-identification protocols, and participants could withdraw at any time without repercussions. The study protocols adhered to institutional and regulatory ethical standards for conducting research involving human subjects.

## 3. Results

Twelve family caregivers (10 females and two males) were enrolled through a public special school in HK. As shown in Table 2, all participants were of Chinese ethnicity. Among their family members with ID, nine had mild ID, while three had moderate ID with accompanying speech and adaptive behavior challenges. The main and subthemes are shown in Table 3.


**Theme 1: Planning a loved one’s future**


Many participants indicated that they postponed or delayed long-term care planning despite occasions when they contemplated their function in supporting their ID family member. They faced obstacles that hindered their ability to make long-term care plans.


**
*Guilt over respite from caring for a disabled child*
**


All participants experienced the discussion as effectively taxing, and the heartache and blameworthiness of abandoning family frequently impeded the dialogue.

“*My youngest brother’s complete trust in me has always made me feel responsible for him. As he gets older, I am wrestling with slowly transferring that duty of care and loosening the apron strings.*” (P10, female, sibling, mild intellectual limitation)

One participant grappled with placing her adult child in residential care for limited days per week. Despite lightening her burden, she thought she must always care for her child without respite. Her child functioned mentally equivalent to a pre-adolescent. She felt duty-bound to protect and care for her ID child ceaselessly. She struggled with shame about occasionally ceding her child’s care, even knowing its benefits. She anguished over whether intermittent residential care implied she fell short as a perpetual caregiver.

“*Reintegrating my son into family life after a prolonged absence from residential care due to the pandemic presents significant challenges. While I strive to be understanding of his readjustment, maintaining reasonable expectations is crucial, given the constraints of our home environment (e.g., considerate neighbors). My primary concerns are preventing regression to previous behaviors. A successful transition requires balancing compassionate support with firm guidance to ensure his progress is maintained before his eventual return to care.*” (P07, female, parent, mild intellectual limitation)


**
*Discouraged by lack of good housing options*
**


Some participants described encountering obstacles in their efforts to secure appropriate housing or residential care, reporting a lack of acceptable options. For numerous participants, the process of locating appropriate accommodations and care had been prolonged, with some still engaged in the search. These participants expressed feelings of discouragement regarding the ongoing housing situation.

“*My son is finishing secondary school, and he enjoys brewing coffee. His school had previously set up some baking training opportunities for him, however he never ended up participating in them. Unfortunately, when the pandemic hit, those plans fell apart before he had the chance to take advantage of the baking training. It was just bad timing; the training aligned with his interests, but external circumstances prevented him from being able to follow through with it … I have struggled to find housing and support programs for him after graduation. There are long waitlists and high costs. I worry he will not be accepted and understood. But my son is kind and has much to offer. Though it will be hard when he leaves school, I will face this challenge with him. I want to help his transition to a happy, safe life in the community.*” (P04, male, parent, moderate intellectual limitation)


**
*Exhaustion from total devotion to a disabled child*
**


The participants reported that the demanding caregiving within the home often monopolized all their effort and concentration, preventing them from focusing on anything else. Caregiving responsibilities significantly impacted family relationships, social connections, employment, and caregivers’ personal well-being. However, when others raised worried about potential fatigue, it could prompt the caregiver to consider dispersing some tasks and obligations to divide the caregiving role with others. This apportioning and handing off duties had the potential to alleviate the primary caregiver’s burden. The feedback from third parties seemed to provide opportunities for critical reflection on the care situation when caregivers were consumed by their role.

“*The pandemic disrupted my son’s post-secondary plans, leaving him without work or further training for a year. Now, I manage his daily activities and social engagements, but this constant support is exhausting. Finding age-appropriate social connections is difficult, and family vacations are challenging. My goal is to help him find fulfillment and, eventually, independence, but for now, we focus on each day.*” (P11, female, parent, moderate intellectual limitation)


**
*Burden of being the only true caregiver*
**


Participants expressed concern regarding the overall well-being of their kin with ID. Their apprehension frequently related to aspects that only the participants seemed to comprehend fully, owing to their profound familiarity with the precise needs of their kin. Most participants feared that nobody besides themselves could provide adequate care. Furthermore, because of experiencing incomprehension among their social circles, some completely severed ties, leaving them with few trusted individuals on whom they could depend. They were also reluctant to overburden their loved ones.

“*I worry daily about caring for my son. I know what he needs, his favorite foods, his comforting routines. I understand him in a way nobody else does. I fear I am the only one who can provide the patient care he requires. When people do not understand his behaviors, I feel isolated. Now I have few I can truly rely on. My other children try to help, but I do not want to burden them too much. They have their own busy lives. I will always be the one my disabled son depends on most. This responsibility weighs on me, but his smile and hugs make it all worthwhile. I could not imagine life without him. I will always do whatever it takes to provide the care he needs.*” (P11, female, parent, moderate intellectual limitation)


**Theme 2: Burden of arranging care for disabled loved ones**


Though in the back of their minds, most participants of family members with ID have long-held questions about future care provision if unable to continue. These issues, typically unspoken, sometimes come to the forefront and require addressing.


**
*Challenges of caring for a disabled sibling*
**


Siblings frequently anticipated inheriting sole or shared caregiving duties for their impaired relations. Considerable apprehension around this issue was recounted extensively, with frequent unresolved examination. Therefore, these brothers and sisters providing care came to see that they might be unable to continue their devotion because it was incompatible with other responsibilities and requirements.

“*As the oldest sibling, I knew caring for my brother with ID would fall to me someday. Now, with my parents aging, that dreaded day looms. Though I adore my sweet brother, providing his constant care is impossible with my own family’s needs. My heart is shattered, knowing I’ll likely have to betray our valued bond, placing him in a care facility. I desperately wish there was another way, but time has run out to find one.*” (P10, female, sibling, mild intellectual limitation)


**
*Difficulty planning a loved one’s future care*
**


The anxieties regarding care continuity commonly coincided with participant aging, deteriorating health, and contemplations of finitude. As they got older, many participants noticed it was getting harder to provide care. Some had lost parents or friends around their age recently.

“*As I have gotten older, it has gotten harder to care for my son, both physically and emotionally. I worry about what will happen to him when I am gone since he depends on me. Although it is difficult, I will treasure our time together and try to plan for his care in the future*.” (P07, female, parent, mild intellectual limitation)

“*My son completed his applied learning program in Tai Po this past July. I was proud to see him graduate. But as I get older, I worry about his future without me. I care for all his needs now. The thought of him confused and alone when I am gone breaks my heart. He relies on me completely. It’s painful, but I need to plan his care after I pass. I wish I could care for him forever. But I must face reality and ensure he will be looked after when I cannot be there. His well-being must be my priority, even though planning is hard*.” (P03, male, parent, mild intellectual limitation)


**Theme 3: Planning child’s future care amid family tensions**


There was anxiety about discussing future planning, though family members also wished to talk. When the topic was broached, others offered assistance, preserving family bonds. Some involved the ID in the decision, realizing the instability. Talking about the future and seeking help allowed openness for joint planning. Many gradually transferred caregiving tasks, growing more comfortable sharing duties. Sharing guardianship with family provided reassurance.


**
*Relief from family support about future care*
**


Participants indicated that upon finally broaching the subject, others expressed willingness to provide assistance. This resulted in a feeling of gratitude among participants regarding the preservation of familial bonds.

“*I was scared to talk about the future. What if my family got angry? But we had to have the conversation. When I brought it up, they surprised me … They wanted to help and support us. I am so thankful I have their love through any challenge ahead*.” (P06, female, parent, mild intellectual limitation)

Few participants described families in which the ID member was included in decisions and actions. One participant illuminated how this inclusive approach prompted a collective awakening regarding the instability of their current domestic circumstances and the wisdom of planning for future arrangements.

“*When my son was diagnosed (as mild ID), I worried for his future. As he grew up, I knew he should help plan his life. I was scared to include him at first. But we had good talks about his dreams and fears. He showed me how capable he is. I am proud of him for shaping his life, even though it’s hard for me. We even toured a residential care home, but the waitlist was so long.*” (P08, female, parent, mild intellectual limitation)


**
*Tension around discussing future care*
**


Participants expressed that they frequently approach discussions of extended preparations with unease, as they are uncertain of the responses they will receive from others. At times, close family members of the differently abled individual, such as brothers or sisters, also hoped to discuss the subject but were unsure how to begin such conversations. This shared yet unarticulated wish for dialogue, paired with the resulting tension within families and ambiguity regarding prospective circumstances, was conveyed by participants.

“*My heart races every time the words ‘future’ or ‘planning’ coming up. I want to talk about it with my daughter about her younger brother, but my throat tightens up at the thought. What if they react with anger or fear? My stomach twists in knots imagining the conversation. I wish I could find the words to express my hopes and anxieties without upsetting anyone. For now, an uneasy dread washes over me whenever the topic arises.*” (P11, female, parent, moderate intellectual limitation)


**
*Peace of mind planning future care together*
**


Participants stated that discussing prospects and seeking help prompted sincerity about their situation and allowed for joint strategic foresight. Numerous participants had steadily transferred caregiving obligations to others, which cultivated trust in allocating care duties. Participants also depicted distributing custodianship responsibilities with family, which conferred composure.

“*When I make arrangements for my child’s future care, it puts my anxious mind at ease. If I do not plan ahead, I feel unsettled, consumed by uncertainty over what may come. Naturally, I worry more for my ID child than my independent adult children who have moved on with their lives. Knowing there will be reliable support gives me comfort, sparing me from fretting over an inability to somebody provide care myself. My heart feels tranquil when provisions are in place, lifting the burden of ceaseless concern for my vulnerable child’s health*.” (P02, female, parent, moderate intellectual limitation)


**Theme 4: Pandemic worsens future caregiving worries**


Participants expressed elevated health concerns and implemented preventative measures to mitigate risks. Intensified caregiving duties during lockdowns led to exhaustion, lack of respite, and feelings that long-term planning was futile given the shortage. Their reports examine pandemic worries, responses, and caregiver burnout for families of ID relatives.


**
*Pandemic worries about future care*
**


Participants conveying elevated levels of concern revealed fears of acquiring COVID-19 and enduring critical illness or fatality.

“*My son’s activities have been reduced in recent years due to the pandemic. In the past, his program organized events like large-scale game booths, trips, and cooking classes. Now there are fewer engagements. This lack of activity has been difficult for him … As someone vulnerable to the COVID-19 pandemic’s threats, having battle breast cancer, I agonized over my son’s well-being should I fall gravely ill, unable to continue caring for her. In some moments, intense anxiety tore through me, envisioning his alone and adrift, my own imminent mortality starkly apparent.*” (P11, female, parent, moderate intellectual limitation)

After the start of the COVID-19 pandemic, caregivers with an ID family member in their household described more worry about long-term arrangements than those whose family members lived elsewhere. However, people with family members in residential care homes also expressed concern about future circumstances.

“*The pandemic has troubled me with worries for the future even more than before. When my beloved’s school closed and his dormitory barred visitors during lockdown, the harsh separation pierced my heart. I ached at being unable to care for him, just as one day I will be too old or frail to be there. My son had to stay at the dormitory with nothing to do, just idling away the days. The months we lost at me daily. As a parent, imagining a time when I cannot protect him fills me with profound sorrow. Our fractured contact was but a glimpse of the inevitable day when I will no longer be beside him. That distressing preview will linger with me always*.” (P07, female, parent, mild intellectual limitation)


**
*Burnout and desperation during the pandemic*
**


Participants expressed deep exhaustion and inability to recover due to the COVID-19 pandemic. They voiced the need for additional assistance, as they could no longer meet the needs of caring for their ID family member. The COVID-19 pandemic has magnified one major obstacle to considering long-term care strategizing; participants asserted the futility of considering weighing other care options given the lack of space in residential care facilities for their children. All of their ID children pressed them to stay home without school or outside training activities during the COVID-19 pandemic. Those children displayed many uncontrolled emotional outbursts at home. Participants predicted more caregiver burnout would lead to higher demand for residential care homes and longer wait times.

“*My daughter lives in a residential care home where she is active during holidays, previously requiring me to take off work to care for her. This created pressure, exacerbated by pandemic disruptions to routines. With limited respite care still today, I hope to find additional support services for occasional breaks while ensuring my daughter enjoys her time off. The pandemic has drained my energy trying to care for my child without relief, as we’ve been trapped at home with no school or activities. The endless emotional outbursts fray my nerves until I have nothing left. I desperately need specialized help, but care home spots are scarce. All I can do is struggle through the agonizing isolation watching my child regress at home. I worry I will soon reach a breaking point, like other overburdened caregivers stretched thin. Our children deserve better than parents with no support options. I need some form of assistance, or I do not know how much more I can handle*.” (P05, female, parent, mild intellectual limitation)


**
*Pandemic precautions to protect a vulnerable loved one*
**


Due to the COVID-19 pandemic, certain caregivers enacted highly pragmatic actions should they experience an adverse event, such as transferring to community isolation facilities consequent to COVID-19 contagion, in efforts toward forestalling the proliferation of the COVID-19 virus and adhering to governmental directives during the specified period.

“*Pre-pandemic, my daughter enjoyed strong social connection. Lockdowns drastically reduced these. My greatest fear during our COVID-19 infection wasn’t my own illness, but the impact on my daughter, especially the isolation in Penny’s Bay (Penny’s Bay Community Isolation Facility). The experience was incredibly distressing, but I endured it to protect her*.” (P12, female, parent, mild intellectual limitation)

## 4. Discussion

The findings of this study explore the experiences of caregivers for children with ID regarding long-term care planning during the COVID-19 pandemic. The risk of not examining this issue now is that when an emergency strikes, decisions will need to be made hastily, potentially overlooking the preferences and requirements of family members and children with ID [18,34,35].

### 4.1. ID Caregiving and COVID-19: Planning for the Future

The COVID-19 pandemic heightened these concerns, as caregivers worried about their own health and the potential impact on their dependents. Some caregivers took proactive steps, such as making legal preparations and arranging for long-term care options, demonstrating resilience in the face of uncertainty. For example, some caregivers, recognizing the potential for disruption, took practical steps like transferring guardianship and making legal arrangements (P12), demonstrating a proactive approach to mitigating future risks. Others, despite the challenges, involved their children with ID in the planning process, fostering a sense of agency and shared decision-making (P08).

While the pandemic spurred increased engagement in planning, it was not always collaborative. Many caregivers made independent decisions, sometimes excluding family members or family members with ID in Theme 3 (Tension around discussing future care). This approach, while potentially intended to shield their relatives from distress [18,34,35], could create feelings of burden and exclusion for those with ID when presented with pre-determined arrangements [18,35]. However, some families successfully navigated these conversations, finding relief and strengthening bonds through open communication about future care in the findings of Theme 3 (Relief from family support about future care). One participant described a positive experience involving her son in planning, leading to productive discussions about his future (P08). This successful intervention emphasizes the importance of recognizing the agency and decision-making capacity of family members with ID, even in complex matters like long-term care. This highlights the potential benefits of inclusive planning, empowering individuals with ID and fostering stronger family connections.

### 4.2. Including Individuals with ID in Pandemic Care Planning

Our study found that some participants did involve loved ones with ID in decision-making, potentially facilitated by the increased intimacy of lockdown situations. This inclusion was facilitated by several factors related to the pandemic, increased proximity and communication, heightened awareness of vulnerability, enhanced understanding of needs and preferences, and shared responsibility in daily routines for their family members with ID. This contrasts with the more common experience of exclusion and highlights an opportunity for future research to explore effective strategies for including family members with ID in planning processes in Theme 3 (Relief from family support about future care). With families spending more time together at home, there were naturally more opportunities for casual conversations about the future. These informal settings might have made it easier to broach sensitive topics like long-term care, which might otherwise be avoided in more structured environments. For instance, one participant (P08) successfully involved their son in visiting a potential residential care home and discussing his preferences, demonstrating a practical example of inclusive planning. Furthermore, some participants proactively engaged other family members in discussions, finding reassurance and alleviating anxiety about future care provisions. This collaborative approach, as noted by some research, fostered a sense of shared responsibility and strengthened family unity (P06). The positive outcomes reported by P06, where family discussions led to increased support and reduced anxiety, illustrate the benefits of collaborative planning. These positive experiences demonstrate the potential for collaborative planning to build resilience and reduce caregiver burden [4,35,36].

### 4.3. Support Needs and Barriers to Care Planning During the Pandemic

A significant barrier identified in this study, echoing previous research, is the lack of trust in existing and future care options [18,34,35]. This distrust discourages caregivers from exploring alternatives. This is compounded by the increased awareness of unforeseen risks, such as pandemics and health crises, highlighting the need for reliable and trustworthy respite care options in the findings of Theme 4 (Pandemic worries about future care) [4,35,36,37,38]. The pandemic exacerbated these concerns, with caregivers experiencing burnout and desperation due to increased caregiving demands and limited access to support services in the findings of Theme 4 (Burnout and desperation during the pandemic). One participant tearfully described the exhaustion and emotional toll of providing constant care during lockdown, highlighting the urgent need for respite care options (P05). Specially, increasing the availability of subsidized respite care services, creating online platforms for sharing caregiving resources, and establishing community-based support networks could address this critical need. Despite these challenges, some caregivers demonstrated remarkable resilience, taking precautions to protect their vulnerable loved ones and enduring difficult situations to ensure their safety (P12) [39]. Participant P12’s proactive measures, such as transferring guardianship and making legal arrangements, exemplify a successful intervention for mitigating risks during a crisis.

### 4.4. Supporting Caregivers and Individuals with ID: Future Directions

The growing population of aging individuals with ID, coupled with shrinking family support networks and staff shortages, indicates the urgent need for comprehensive support for caregivers and their children [18,34,35]. This includes expanding day programs for individuals with ID and temporary care facilities to alleviate caregiver burnout. For example, implementing voucher programs for respite care, developing specialized training for respite care providers, and creating accessible online directories of respite services could help address caregiver burnout. It also requires facilitating early and open discussions about future residential care and end-of-life preferences, acknowledging the emotional complexities of these conversations in the findings of Theme 1 (Guilt over respite from caring for a disabled child) [39]. Furthermore, addressing the lack of trust in care options is crucial. This involves not only improving the quality and availability of services but also providing caregivers with the information and support they need to make informed decisions in the finding of Theme 1 (Discouraged by lack of good housing options). By acknowledging the challenges, celebrating the successes, and focusing on practical solutions, we can empower caregivers to build sustainable support systems and ensure the long-term well-being of family members with ID. This study reinforces the need for professionals to proactively initiate conversations about long-term care planning, recognizing the emotional burden on caregivers and offering guidance and support. For example, professionals can offer guidance on how to initiate conversations, provide resources on person-centered planning, and facilitate communication between family members and the individual with ID. Specially, interventions such as counseling, support groups, and educational resources can help caregivers process their emotions, connect with others facing similar challenges, and develop coping strategies [9,40,41]. Validation of their feelings of guilt, grief, and anxiety is crucial. Furthermore, providing clear information about available care options and resources can alleviate fears and empower caregivers to make informed decisions that align with their values and priorities [41]. Respite care services can also provide temporary relief from caregiving responsibilities, reducing burnout and creating space for emotional processing and planning. This proactive approach can empower caregivers to navigate the complexities of planning, fostering a sense of hope and resilience in the face of uncertainty [42].

### 4.5. Limitations

The small sample size limits extending conclusions to all HK caregivers for children with ID. Purposive sampling may introduce bias as participants self-selected during the COVID-19 pandemic. Findings may not apply globally, since needs likely vary across differing socioeconomic conditions and disability services.

### 4.6. Implications

The implications of this study recommended that HK care purveyors confer with family members tending to intellectually impaired loved ones, implement long-term care strategizing, and furnish continuing assistance to amplify the availability of specialized health provisions for their disabled family member. The welfare of caregivers and their caregiving duties necessitate prompt measures as they seek backing to cope with the pandemic’s fallout and lengthy waitlists for residential care homes for their intellectually disabled children.

## 5. Conclusions

This qualitative study explored HK caregivers’ perspectives on long-term planning for relatives with ID during the later period of the COVID-19 pandemic. Interviews revealed key triggers raising awareness of future planning needs, but barriers like painful emotional distress and exhaustion hindered progress; open communication and family involvement helped. The pandemic exacerbated challenges and fragile support networks. Healthcare providers must offer individualized psychosocial support, practical assistance and skill-building workshops, increased respite care, and improved care coordination. Policymakers must prioritize the gradual implementation of supportive policies, increased investment in services, facilitated discussions about resident care, and a comprehensive, integrated care plan. Addressing caregiver needs requires a collaborative effort to build more resilient support systems, improving the quality of life for both caregivers and their relatives with ID.

## Figures and Tables

**Table 1 healthcare-12-02512-t001:** Synopsis of techniques used to attain methodological rigor.

Criteria	Purpose	Approaches to Achieve Rigor	Remarks on Implementation
Credibility	To demonstrate that the data is accurate, reputable, and plausible	Prolonged and diverse acquaintance with environmental milieu	- Telephone, Zoom, or in-person interviews were conducted but only because of the personal choice of the participants. - Nonetheless, various participants across several special schools were contacted through social workers who maintained ongoing dialogue about this research with the authors’ affiliated organization.
		The approach and strategies utilized during interviews	- After obtaining ethical approval, the interview guide questions were trailed at one introductory session. The first interview was then conducted via in-person, with data from these interviews incorporated into the final data analysis.
		Gathering of benchmark data for comparison	- Field notes were utilized to support documentation of contextual information cited by participants for precise data analysis. Field notes were also examined alongside the transcripts.
		Validating the investigators’ credentials	- The entire research team held the requisite understanding, data administration abilities, and hands-on qualitative research practice of 3 or more years to execute their functions.
		Peer review	- Regular debriefing discussions every 2 weeks with the Hong Kong Academy of Nursing and Midwifery Fellows ensured no inadvertent prejudices, standpoints, or suppositions were made by the researchers.
Dependability	To confirm the results of this qualitative research would be reproducible if the exploration transpired again amongst the same population	A comprehensive account of research procedures	- The study techniques were comprehensively illustrated and unambiguously expressed in the scholarly articles.
		Instituting a reviewable record	- All investigators constructed a meticulous audit trail of the data-gathering procedure. - Member check was conducted to guarantee the clarity of the meanings obtained from participants, thus improving the validity of participant narratives.
		Gradual reproduction of information in stages	- The entire research team evaluated encoding precision and cross-checker dependability over the course of the data-parsing process.
Confirmability	To broaden the assurance that the findings would be validated/substantiated by other investigators	Self-reflection	- Practices like reflective diaries and biweekly scholar meetings were embraced.
Transferability	To broaden the extent to which the findings can be abstracted/conveyed to other situations	Elaborate representation and data saturation	- Comprehensive elaboration was submitted during the selection of the stakeholders; therefore, the significance of the expressions from the affiliates could be understood amid the context. - Data completeness was attained when further concepts surfaced from the subjects. All investigators concurred on the achievement of evidence saturation.

**Table 2 healthcare-12-02512-t002:** Characteristics of participants.

Characteristics		*N* = 12	
		*n*	%
Sex			
	Male	2	17
	Female	10	83
Relationship with ID individual			
	Parent	11	92
	Sibling	1	8
Family member’s intellectual limitations			
	Mild	9	75
	Moderate	3	25

**Table 3 healthcare-12-02512-t003:** Themes and subthemes of this study.

Themes	Subthemes
Planning a loved one’s future alone	- Guilt over respite from caring for a disabled child
	- Discouraged by lack of good housing options
	- Exhaustion from total devotion to a disabled child
	- The burden of being the only true caregiver
Burden arranging care for disabled loved ones	- Challenges of caring for a disabled sibling
	- Difficulty planning a loved one’s future care
Planning child’s future care amid family tensions	- Relief from family support about future care
	- Tension around discussing future care
	- Peace of mind planning future care together
Pandemic worsens future caregiving worries	- Pandemic worries about future care
	- Burnout and desperation during the pandemic
	- Pandemic precautions to protect a vulnerable loved one

## Data Availability

The data that support the findings of this study are available from the corresponding author upon reasonable request.

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
