# Peer review of "Navigating a New Normal: A Qualitative Look at Long-Term Care Planning for Children with Intellectual Disabilities Post-COVID-19"

_healthcare, 2024, doi:10.3390/healthcare12242512_

Round 1
Reviewer 1 Report
Comments and Suggestions for Authors
This paper addresses a critical and underexplored area of research: the challenges faced by family caregivers of individuals with intellectual disabilities (ID) in long-term planning, particularly during the later stages of the COVID-19 pandemic. The study's focus on Hong Kong caregivers brings a valuable cultural and regional perspective to the global discourse on caregiving and planning for individuals with disabilities.
Using a qualitative phenomenological approach and in-depth interviews analyzed via Colaizzi's method lends depth and credibility to the findings. The identification of four distinct themes—addressing personal, familial, and societal challenges—provides a comprehensive framework for understanding caregivers' experiences. Furthermore, the implications for policy and practice are thoughtfully articulated, emphasizing the need for targeted support, improved resources, and facilitated planning discussions.
The study is well-positioned to fill a gap in the literature, and its insights are timely and actionable. They offer a foundation for future research and inform interventions that could significantly improve caregiver support systems. However, it may benefit from additional details on the sampling rationale, the broader applicability of findings, and any potential limitations in data collection or analysis.
Suggestions for Results:
Some sections, mainly participant quotes, are overly lengthy. Condensing these could improve readability without losing meaning.
While the quotes are rich, there could be more synthesis. Explicitly summarizing how the quotes support or exemplify the themes would strengthen the connection between the data and the findings.
Repetitive elements (e.g., multiple mentions of guilt over using respite care) might distract readers. Streamlining these could make the findings more focused.
Some sentences are complex or grammatically awkward (e.g., "encumbered their bonds, family life, camera derie, work, and personal health"). Ensuring grammatical accuracy and avoiding typos (e.g., camera derie should likely be camaraderie) will enhance professional presentation.
Suggestions for discussion:
The discussion section contains many ideas, but the organization could be improved for clarity and readability. Consider grouping ideas under clear subheadings. This would help readers follow the argument and key findings more easily.
Balance Positive and Negative Findings, for instance, highlight examples of success stories or practical steps caregivers have taken during the pandemic.
The text mentions that some caregivers included family members with ID in planning due to closer living arrangements during the pandemic. Elaborate on this:
- What specific factors facilitated this inclusion?
- How did this impact the caregivers and the individuals with ID?
Discuss how these insights could inform future strategies for more inclusive planning.
While the section identifies several challenges, the practical solutions could be more specific. For example:
Instead of “increasing respite options,” specify what programs (e.g., day programs for individuals with ID, temporary care facilities) could be most impactful.
Provide examples of successful interventions from these studies.
How can professionals effectively engage individuals with ID in long-term planning without causing distress?
What interventions can help caregivers overcome the emotional barriers to planning?
Phrases such as "sustain their health, maintain family relationships, and actively participate in planning" appear multiple times. Condense for conciseness.
Replace overly complex phrasing with simpler alternatives (e.g., "the ominous shadow of COVID-19" can be rephrased as "the challenges brought by COVID-19").
Author Response
Dear Respected Reviewer,
Healthcare
RE: Manuscript titled “Navigating a New Normal: Long-term Care Planning for
Children with Intellectual Disabilities After COVID-19”
Thank you for your unstinting effort to review the revised manuscript. Our research team appreciated your valuable comments. In this document, we provided our responses in a point-by-point format. We hope that our revisions to the manuscript can address your concerns satisfactorily.
This response letter was prepared with reference to a template that we downloaded from the Journal website. In that template, we noticed that all responses from the authors were highlighted in red. We thus followed the prescribed style and format.
Yours faithfully,
Corresponding author
Manuscript no.: healthcare-3314607
Comments 1: Are the results clearly presented (x) can be improved
Response 1: Thank you for pointing this out. We agree with this comment, Therefore, we have revised the results presentation in page 7 line 237 to page 11 line 459.
Comments 2: Are the conclusions supported by the results (x) can be improved
Response 2: Thank you for pointing this out. We agree with this comment, Therefore, we have revised the results presentation in page 13 line 580 to page 14 line 591.
“This qualitative study explored HK caregivers’ perspectives on long-term planning for relatives with ID during the later period of COVID-19 pandemic. Interviews revealed key triggers raising awareness of future planning needs, but barriers like painful emotional distress and exhaustion hindered progress; open communication and family involvement helped. The pandemic exacerbated challenges and fragile support networks. Healthcare providers must offer individualized psychosocial support, practical assistance and skill-building workshops, increased respite care, improved care coordination. Policymakers must prioritize gradual implementation of supportive policies, increased investment in services, facilitated discussions about resident care, and a comprehensive, integrated care plan. Addressing caregiver needs requires a collaborative effort to build more resilient support systems improving quality of life for both caregivers and their family members with ID.”
Comments 3: Some sections, mainly participant quotes, are overly lengthy. Condensing these could improve readability without losing meaning. While the quotes are rich, there could be more synthesis. Explicitly summarizing how the quotes support or exemplify the themes would strengthen the connection between the data and the findings.
Response 3: Thank you for pointing this out. We agree with this comment. Therefore, we modified the results section in page 7 line 255 to 261:
“Reintegrating my son into family life after a prolonged absence from residential care due to the pandemic presents significant challenges. While I strive to be understanding of his readjustment, maintaining reasonable expectations is crucial, given the constraints of our home environment (e.g., considerate neighbors). My primary concerns are preventing regression to previous behaviors. A successful transition requires balancing compassionate support with firm guidance to ensure his progress is maintained before his eventual return to care.” (P07, female, parent, mild intellectual limitation)
In page 8 line 288 to 293:
“The pandemic disrupted my son’s post-secondary plans, leaving him without work or further training for a year. Now, I manage his daily activities and social engagements, but this constant support is exhausting. Finding age-appropriate social connections is difficult, and family vacations are challenging. My goal is to help him find fulfillment and, eventually, independence, but for now, we focus on each day.” (P11, female, parent, moderate intellectual limitation)
In page 11 line 455 to 459:
“Pre-pandemic, my daughter enjoyed strong social connection. Lockdowns drastically reduced these. My greatest fear during our COVID-19 infection wasn’t my own illness, but the impact on my daughter, especially the isolation in Penny’s Bay (Penny’s Bay Community Isolation Facility). The experience was incredibly distressing, but I endured it to protect her.” (P12, female, parent, mild intellectual limitation)
The analysis of caregiver experiences and challenges related to planning for loved ones with ID can be significantly strengthened by explicitly linking the provided quotes to the identified themes. While the themes of future planning obstacles, caregiving burdens, family tensions surrounding care discussions, and pandemic-exacerbated worries are relevant, the analysis currently relies on implicit connections between the data and findings. A more robust analysis would explicitly demonstrate how each quote exemplifies the corresponding theme or subtheme, strengthening the connection between the lived experiences of caregivers and the study’s conclusions. For example, instead of simply presenting a quote about a caregiver’s exhaustion, the analysis should directly explain how the quote’s content reflects the theme of exhaustion. This explicit linking will enhance the rigor and clarity of the findings, making the study more compelling and persuasive.
Comments 4: Repetitive elements (e.g., multiple mentions of guilt over using respite care) might distract readers. Streamlining these could make the findings more focused.
Response 4: Thank you for your insightful comment regarding the repetitive elements in the findings. We agree that streamlining these instances would create a more focused presentation. In the revised manuscript, we have consolidated the discussions of guilt surrounding respite care. Instead of mentioning guilt repeatedly within each theme, we have now primarily addressed this feeling within the subtheme ‘Guilt over respite from caring for a disabled child’ under Theme 1: Planning a loved one’s future. This consolidation allows for a more in-depth exploration of the guilt experienced by caregivers while avoiding unnecessary repetition throughout the manuscript. We believe this revision creates a more cohesive and impactful narrative, allowing the other important themes and subthemes to stand out more clearly.
Comments 5: Some sentences are complex or grammatically awkward (e.g., “encumbered their bonds, family life, camera derie, work, and personal health”). Ensuring grammatical accuracy and avoiding typos (e.g., camera derie should likely be camaraderie) will enhance professional presentation.
Response 5: Agree. We have, accordingly, modified this point in page 8 line 281 to 282:
“Caregiving responsibilities significantly impacted family relationships, social
connections, employment, and caregivers’ personal well-being.”
Comments 6: The discussion section contains many ideas, but the organization could be improved for clarity and readability. Consider grouping ideas under clear subheadings. This would help readers follow the argument and key findings more easily.
Response 6: Thank you for pointing this out. We agree with this comment. Therefore, we have revised subheadings in the discussion section, in page 11 line 469, page 12 line 493 and 516, and page 13 line 535.
Comments 7: Balance Positive and Negative Findings, for instance, highlight examples of success stories or practical steps caregivers have taken during the pandemic.
Response 7: Thank you for pointing this out. We revised the content of discussion section in page 12 line 505 to page 13 line 558.
Comments 8: The text mentions that some caregivers included family members with ID in planning due to closer living arrangements during the pandemic. Elaborate on this:
Response 8: Thank you for your suggestion. We elaborated in page 12 line 495-499.
“This inclusion was facilitated by several factors related to pandemic, increased proximity and communication, heightened awareness of vulnerability, enhanced understanding of needs and preferences, and shared responsibility in daily routines for their family members with ID.”
Comments 9: What specific factors facilitated this inclusion? How did this impact the caregivers and the individuals with ID? Discuss how these insights could inform future strategies for more inclusive planning.
Response 9: Thank you for your suggestion. We elaborated in page 12 line 503-514.
Comments 10: While the section identifies several challenges, the practical solutions could be more specific. For example:
Response 10: Agree. We modified the content be more specific in page 12 line 526-529, page 13 line 539-542.
“Specially, increasing the availability of subsidized respite care services, creating online platforms for sharing caregiving resources, and establishing community-based support networks could address this critical need.”
“For example, implementing voucher programs for respite care, developing specialized training for respite care providers, and creating accessible online directories of respite services could help address caregiver burnout.”
Comments 11: Instead of “increasing respite options,” specify what programs (e.g., day programs for individuals with ID, temporary care facilities) could be most impactful.
Response 11: Agree. We revised the content in page 13 line 538-539.
“This includes expanding day programs for individuals with ID, and temporary care facilities, to alleviate caregiver burnout.”
Comments 12: Provide examples of successful interventions from these studies.
Response 12: Thank you for your suggestion. We elaborated more with examples in page 12 line 487-489, 505-507, 511-513, 531-533.
“This successful intervention emphases the importance of recognizing the agency and decision-making capacity of family members with ID, even in complex matters like long-term care.”
“For instance, participant (P08) successfully involved their son in visiting a potential residential care home and discussing his preferences, demonstrating a practical example of inclusive planning.”
“The positive outcomes reported by P06, where family discussions led to increased support and reduced anxiety, illustrate the benefits of collaborative planning.”
“Participant P12’s proactive measures, such as transferring guardianship and making legal arrangements, exemplify a successful intervention for mitigating risks during a crisis.”
Comments 13: How can professionals effectively engage individuals with ID in long-term planning without causing distress?
Response 13: Thank you for pointing this out. We agree with this comment in page 13 line 553-556, 563-565.
“For example, professionals can offer guidance on how to initiate conversations, provide resources on person-centered planning, and facilitate communication between family members and the individual with ID.”
“This proactive approach can empower caregivers to navigate the complexities of planning, fostering a sense of hope and resilience in the face of uncertainty [42].”
Comments 14: What interventions can help caregivers overcome the emotional barriers to planning?
Response 14: Agree. We modified the content of the discussion section in page 13 line 556-563.
“Specially, interventions such as counseling, support groups, and educational resources can help caregivers process their emotions, connect with others facing similar challenges, and develop coping strategies [9,40,41]. Validation of their feelings of guilt, grief, and anxiety is crucial. Furthermore, providing clear information about available care options and resources can alleviate fears and empower caregivers to make in-formed decisions that align with their values and priorities [41]. Respite care services can also provide temporary relief from caregiving responsibilities, reducing burnout and creating space for emotional processing and planning.”
Comments 15: Phrases such as “sustain their health, maintain family relationships, and actively participate in planning” appear multiple times. Condense for conciseness.
Response 15: Thank you. Revised the discussion section page 11 line 462 to page 13 line 565.
Comments 16: Replace overly complex phrasing with simpler alternatives (e.g., “the ominous shadow of COVID-19” can be rephrased as “the challenges brought by COVID-19”).
Response 16: Thank you. In the modified discussion section, no wordings as “the ominous shadow of COVID-19”.

Reviewer 2 Report
Comments and Suggestions for Authors
Dear authors, the qualitative study presented here addresses an interesting topic on the return to normality in a population of children with intellectual disabilities.
In my opinion, the title should include some word that identifies the study as qualitative. Is it possible to reduce the number of words in the title?
Keywords should (as far as possible) use MeSH terms. Consider changing "family cares" to "caregivers". I propose to include "Qualitative research" as keyword.
Introduction: In my opinion it is very extensive (On the contrary, there is very little discussion of the results; only 4 new references in the discussion). Could it be reduced as much as possible? Check the use of the acronym HK for Hong Kong. In the abstract this acronym was used, but in the text it has never been used. The acronym HK should be clarified in its first use in the text of the manuscript and then the acronym should be used the rest of the times it appears.
The aim in the abstract is not described in the same way as in the manuscript. The aim should be unified.
Methods: Ethical aspects should not be included under the sub-heading "study design". Move the paragraph on ethical criteria to another section at the end of the methodology.
The sub-heading participants raises questions about the inclusion criteria when describing that family caregivers of adults with ID were included.... The title indicates Children with ID. Clarify the inclusion criteria for caregivers.
Were more participants contacted than were eventually included in the study, and describe the reasons why they did not participate finally?
Living in HK (iii) is a redundant criterion as it is repeated in relation to criterion (i). Exclusion criteria should be indicated or describe that they have not been considered exclusion criteria (if applicable).
A sub-heading on methodological rigour should be included (move the current information on rigour in point 2.4 and Table 1). A sub-heading on researcher characteristics and reflexivity (COREQ Domain 1) should also be included.
The interview guide should be provided as a supplementary file and quoted in the text in section 2.3 Data. (It is recommended to change Data to "access to participants".
The process of categorisation (intentional vs. emergent?) and the process of coding verbatims as units of meaning into codes, sub-themes and themes should be further clarified in section 2.4.
Transcriptions were provided to participants for verification and approval? describe these aspects in 2.4 (or in the point of rigour).
Results: Table 2 (with frequencies and percentages) is not suitable for representing results on socio-demographic characteristics in qualitative studies. Each of the 12 participants should be specified with their characteristics; for example: P10 female, parent, Intellectual limitation Moderate, ethnicity Chinese. Have other characteristics of interest such as age, occupation, education not been considered?
Table 2 and table 3 should follow immediately after being cited in the text. For table 3, some information must be provided in the text.
It is not correct that subheading 3.1 corresponds to theme 1. The results begin with sociodemographic characteristics and the thematic analysis as a second point. Remove the subheadings (3.1., 3.2,...) and 3.1.1, 3.1.2,... in the results.
Author Response
Dear Respected Reviewer,
Healthcare
RE: Manuscript titled “Navigating a New Normal: Long-term Care Planning for Children with Intellectual Disabilities After COVID-19”
Thank you for your unstinting effort to review the revised manuscript. Our research team appreciated your valuable comments. In this document, we provided our responses in a point-by-point format. We hope that our revisions to the manuscript can address your concerns satisfactorily.
This response letter was prepared with reference to a template that we downloaded from the Journal website. In that template, we noticed that all responses from the authors were highlighted in red. We thus followed the prescribed style and format.
Yours faithfully,
Corresponding author
Manuscript no.: healthcare-3314607
Comments 1: Does the introduction provide sufficient background and include all relevant references (x) can be improved
Response 1: Thank you for pointing this out. We agree with this comment, Therefore, we have revised the relevant references in page 1 line 32 to page 3 line 143.
Comments 2: Is the research design appropriate (x) can be improved
Response 2: Following your feedback, we revised the analysis strategy and rigor of research design (page 4 line 168 to page 5 line 218).
Comments 3: Are the methods adequately described (x) can be improved
Response 3: Thanks for bringing this to our attention. We’ve updated the analysis strategy and rigor of research methods on page 4 line 168 to page 5 line 218.
Comments 4: Are the results clearly presented (x) can be improved
Response 4: Thank you for pointing this out. Discuss the changes made, we updated the results in page 7 line 237 to page 11 line 458.
Comments 5: Are the conclusions supported by the results (x) can be improved
Response 5: Agree. We revised the content of the conclusion section in page 13 line 580 to page 14 line 592.
Comments 6: the title should include some word that identifies the study as qualitative. Is it possible to reduce the number of words in the title?
Response 6: Agree. We have, accordingly, modified this point in page 1 line 2 to 4:
“Navigating a New Normal: A Qualitative Look at Long-term Care Planning for Children with Intellectual Disabilities Post-COVID-19”
Comments 7: Keywords should (as far as possible) use MeSH terms. Consider changing "family cares" to "caregivers". I propose to include "Qualitative research" as keyword.
Response 7: Thank you for pointing this out. We agree with this comment. Therefore, we have revised keywords in page 1 line 29:
“COVID-19, family cares, intellectual disabilities, long-term care, qualitative research”
Comments 8: Introduction: In my opinion it is very extensive (On the contrary, there is very little discussion of the results; only 4 new references in the discussion). Could it be reduced as much as possible? Check the use of the acronym HK for Hong Kong. In the abstract this acronym was used, but in the text it has never been used. The acronym HK should be clarified in its first use in the text of the manuscript and then the acronym should be used the rest of the times it appears.
Response 8: Agree. We have, accordingly, modified all references in introduction and discussion section in page 1 line 36 to page 4 line 155. Moreover, the acronym HK for Hong Kong clarified in its first use in the text of the manuscript and then the acronym used the rest of the times it appears.
Comments 9: The aim in the abstract is not described in the same way as in the manuscript. The aim should be unified.
Response 9: Agree. We have modified this point in page 4 line 154 to 155:
“The investigation aims to explore HK caregivers’ perspectives on long-term planning for family members with ID at the later period of the pandemic.”
Comments 10: Methods: Ethical aspects should not be included under the sub-heading "study design". Move the paragraph on ethical criteria to another section at the end of the methodology.
Response 10: Thank you for pointing this out. Therefore, we revised to another subheading for ethical aspects in page 5 line 219 to line 226.
Comments 11: The sub-heading participants raises questions about the inclusion criteria when describing that family caregivers of adults with ID were included.... The title indicates Children with ID. Clarify the inclusion criteria for caregivers.
Response 11: Thank you for your comment to clarify the inclusion criteria. We modified in page 4 line 168 to 171.
Comments 12: Were more participants contacted than were eventually included in the study, and describe the reasons why they did not participate finally?
Response 12: We contacted only those participants who were ultimately included in the study.
Comments 13: Living in HK (iii) is a redundant criterion as it is repeated in relation to criterion (i). Exclusion criteria should be indicated or describe that they have not been considered exclusion criteria (if applicable).
Response 13: Thank you for your comment. We revised the inclusion criteria in page 4 line 168 to 171.
Comments 14: A sub-heading on methodological rigour should be included (move the current information on rigour in point 2.4 and Table 1). A sub-heading on researcher characteristics and reflexivity (COREQ Domain 1) should also be included.
Response 14: Agreed. We revised the content and added sub-heading on methodological rigour in page 4 line 172 to page 5 line 209.
Comments 15: The interview guide should be provided as a supplementary file and quoted in the text in section 2.3 Data. (It is recommended to change Data to "access to participants".
Response 15: Thank you. We already uploaded the interview questions and stated in the content in page 4 line 172, “2.3. Access to participants”
Comments 16: The process of categorisation (intentional vs. emergent?) and the process of coding verbatims as units of meaning into codes, sub-themes and themes should be further clarified in section 2.4.
Response 16: The revised content for the process of categorisation and the process of coding verbatims in section 2.4. in page 4 line 197 to page 5 line 208.
Comments 17: Transcriptions were provided to participants for verification and approval? describe these aspects in 2.4 (or in the point of rigour).
Response 17: Thank you with this comment. We change this content in page 4 line 197 to page 5 line 218.
Comments 18: Results: Table 2 (with frequencies and percentages) is not suitable for representing results on socio-demographic characteristics in qualitative studies. Each of the 12 participants should be specified with their characteristics; for example: P10 female, parent, Intellectual limitation Moderate, ethnicity Chinese. Have other characteristics of interest such as age, occupation, education not been considered?
Response 18: Thank you pointing this out. We modified participants’ characteristics after each quote according to your comments in page 7 line 248 to page 11 line 459.
Comments 19: Table 2 and table 3 should follow immediately after being cited in the text. For table 3, some information must be provided in the text.
Response 19: Thank you pointing this out. According to your comments, we revised the results in page 7 line 238 to page 11 line 459.
Comments 20: It is not correct that subheading 3.1 corresponds to theme 1. The results begin with sociodemographic characteristics and the thematic analysis as a second point. Remove the subheadings (3.1., 3.2, ...) and 3.1.1, 3.1.2, ... in the results.
Response 20: Agree. We remove the subheadings (3.1., 3.2. … 3.1.1., 3.1.2.) of the
Results section in page 7 line 238 to page 11 line 459.
